# Insight into Recent Progress and Perspectives in Improvement of Antioxidant Machinery upon PGPR Augmentation in Plants under Drought Stress: A Review

**DOI:** 10.3390/antiox11091763

**Published:** 2022-09-07

**Authors:** Hittanahallikoppal Gajendramurthy Gowtham, Sudarshana Brijesh Singh, Natarajamurthy Shilpa, Mohammed Aiyaz, Kalegowda Nataraj, Arakere Chunchegowda Udayashankar, Kestur Nagaraj Amruthesh, Mahadevamurthy Murali, Peter Poczai, Abdul Gafur, Waleed Hassan Almalki, R. Z. Sayyed

**Affiliations:** 1Department of PG Studies in Biotechnology, Government Science College (Autonomous), Nrupathunga Road, Bangalore 560001, India; 2Department of Studies in Botany, University of Mysore, Manasagangotri, Mysuru 570006, India; 3Department of Studies in Microbiology, University of Mysore, Manasagangotri, Mysuru 570006, India; 4Department of Studies in Biotechnology, University of Mysore, Manasagangotri, Mysuru 570006, India; 5Finnish Museum of Natural History, University of Helsinki, 00100 Helsinki, Finland; 6Sinarmas Forestry Corporate Research and Development, Perawang 28772, Indonesia; 7Department of Pharmacology, College of Pharmacy, Umm Al Qura University, Makkah 77207, Saudi Arabia; 8Department of Microbiology, PSGVP Mandal’s, S.I. Patil Arts, G.B. Patel Science & STKV Sangh Commerce College, Shahada 425409, India

**Keywords:** antioxidants, drought, induced systemic tolerance, phytohormones, plant growth promoting rhizobacteria

## Abstract

Agriculture has a lot of responsibility as the rise in the world’s population demands more food requirements. However, more than one type of biotic and abiotic stress continually impacts agricultural productivity. Drought stress is a major abiotic stress that significantly affects agricultural productivity every year as the plants undergo several morphological, biochemical, and physiological modifications, such as repressed root and shoot growth, reduced photosynthesis and transpiration rate, excessive production of reactive oxygen species (ROS), osmotic adjustments, and modified leaf senescence regulating and stress signaling pathways. Such modifications may permanently damage the plants; therefore, mitigation strategies must be developed. The use of drought resistant crop cultivars is more expensive and labor-intensive with few advantages. However, exploiting plant growth promoting rhizobacteria (PGPR) is a proven alternative with numerous direct and indirect advantages. The PGPR confers induced systemic tolerance (IST) mechanisms in plants in response to drought stress via multiple mechanisms, including the alteration of root architecture, maintenance of high relative water content, improvement of photosynthesis rate, production of phytohormones, exopolysaccharides, ACC deaminase, carotenoids and volatiles, induction of antioxidant defense system, and alteration in stress-responsive gene expression. The commercial application of PGPR as bioinoculants or biostimulants will remain contingent on more robust strain selection and performance under unfavorable environmental conditions. This review highlights the possible mechanisms of PGPR by activating the plant adaptive defense systems for enhancing drought tolerance and improving overall growth and yield.

## 1. Introduction

The world population is expected to reach approximately 9 billion people in 2050 with more than 65% of people solely relying on agriculture to ensure their survival and explore global food security [1,2]. This figure is projected to rise above 90% in developing countries. As a result, agriculture will play a major role in a country’s economy, as well as its food supply. However, conventional agricultural practices face many problems, including the problem of small and fragmented land holdings, lack of sufficient irrigation systems and appropriate agricultural mechanization, unavailability of high-quality seeds, and excessive use of agrochemicals [3]. These issues contribute to soil contamination and a gradual increase in soil erosion, eventually leading to different natural disasters. In addition to the aforementioned problems, many crops are subjected to various biotic and abiotic stresses. Biotic stress, generally caused by several living organisms (such as bacteria, fungi, viruses, parasites, and insects), is a major economic damaging factor in pre-harvest and post-harvest losses of major food crops [4,5,6]. The plant never develops sophisticated adaptive immunity even after repeated biotic stress exposures. Abiotic stresses (such as water logging, drought, soil salinity, unfavorable temperatures, nutrient deficiency, heavy metal toxicity, etc.) in the environment can cause permanent agricultural damage, such as stunted plant growth, reduced crop yield, hampered metabolism, and changes in genetic behavior, which can lead to higher mutation rate in the progeny [7,8,9].

A drought is an instance of major abiotic stress encountered by the plants throughout their life cycle due to inadequate water availability. This stress lowers crop development and yield by negatively affecting the quality of morphological, physiological, and biochemical attributes [10,11]. Hence, it is critical to undertake a real paradigm shift towards agricultural sustainability and to solve water shortage and food security problems. To meet ever-increasing global food requirements, it is necessary to develop effective eco-friendly strategies that can augment the levels of plant endurance during water-deficits and improve crop growth and production [12]. Developing drought-resistant crops, shifting crop cultivation, and establishing resource management practices are some strategies to cope with drought stress [10,13]. However, most of them have disadvantages, such as laborious, time-consuming, cross-contamination and loss of beneficial host plant characteristics. These encourage the utilization of plant growth promoting rhizobacteria (PGPR) as cost-effective, a less laborious and promising approach for providing additional drought stress tolerance benefits along with improved plant performance [9,14,15,16]. Therefore, they have often been considered essential tools for facilitating sustainable agriculture. The present review mainly focuses on the current knowledge concerning the various mechanisms of PGPR to counteract the negative impacts of drought stress on host plants and enhance agricultural productivity in dry lands.

## 2. Plant Growth Promoting Rhizobacteria Mediated Drought Stress Tolerance in Plants

One of the plant’s primary defense responses subjected to drought is the stomatal closure, which reduces the uptake and fixation of carbon dioxide (CO_2_), and photosynthesis, thus potentially causing whole plant death in severe drought stress. The plants use efficient adaptive mechanisms to better cope with drought stress, including the changes in shoot and root morphology, reactive oxygen species (ROS) production, synthesis of stress hormones and activating antioxidant defense system [9]. The significant socioeconomic impact of drought stress on agriculture also results in huge monetary losses as it is incurred with long-term effects. The PGPR are the natural habitants of the rhizosphere soil that rely on root exudates to fuel their metabolic activities [17,18,19]. The great diversity of PGPR is influenced by soil physiological conditions and nutrient availability in the rhizosphere [20,21]. These PGPR provide the plants with various growth promoting benefits, primarily through enabling root colonization. These bacteria act as biofertilizers and biostimulants that are most beneficial for agricultural sustainability by promoting the overall plant growth and yield directly or indirectly under severe drought stress [12,22,23]. The most prominent mechanisms of PGPR employed for mitigating drought stress (Figure 1) are elaborated below.

### 2.1. Alteration of Host Root System Architecture

The plant phenotypic response to water stress was mainly associated with the alteration of their root system architecture due to the production of ROS. One of the first steps toward improved drought stress tolerance is the bacterial-mediated alteration in host root system architecture [24,25]. The PGPR candidates of *Bacillus* sp. and *Enterobacter* sp. displayed alteration of root system architecture in *Triticum aestivum* and *Zea mays* that facilitated the drought avoidance phenotypes viz., enhanced root surface area, root branching, root length, and many root tips when compared to the untreated control plants [26]. Many studies have revealed that the PGPR enhance plant cell membrane stability and root nodulation by modulating the level of phytohormones and activating the antioxidant defense mechanism that is directly correlated with increasing plant drought tolerance [27,28]. These research findings implied the possible role of PGPR in mitigating plant stress tolerance through altering their root architecture.

### 2.2. Maintenance of Relative Water Content

The relative water content is a useful indicator of leaf water status in the plant as it is involved in plant metabolism. The drought causes a fall in relative water content, often followed by a reduction in transpiration rate and leaf water potential [29]. The turgor pressure also decreases in the plant cells when water levels fall, which consequently causes cell damage, wilting, and a decline in plant development [30]. The drought usually causes a decrease in hydraulic conductivity, which alters the relative water content of the plant. Drought-induced ROS and osmotic stress are thwarted by the plants’ high relative water content, which may increase crop productivity [1,4,31]. As a result, relative water content measurements are crucial for determining the severity of drought stress. The PGPR regulates water potential by modifying the hydraulic conductivity, stomatal opening and transpiration rate that improve the plant survival rate in drought stress conditions [32]. The literature indicated that the PGPR treatment effectively caused the root system development that improves the water uptake, enabling inoculated plants to tolerate drought stress [24,33]. The exact mechanisms underlying the increased relative water content with PGPR application during drought stress have yet to be explored. However, it is believed that PGPR can help the plant increase its relative water content due to bacterial production of phytohormones and osmoprotectants and alleviate the drought stress [34]. It has been noted that inoculation of *T. aestivum* plants with *Azospirillum* sp. significantly increased the relative water content attributed to the bacterial IAA production. Apart from increasing the relative water content, the bacterial treatment also enhanced root growth and the formation of lateral roots, which help uptake water and mineral nutrients under drought stress [19]. Similarly, *Arabidopsis* plants inoculated with ABA producing *A. brasilense* improved the relative leaf water content, thereby ameliorating growth response under drought stress [22], which confirmed the ability of phytohormone producing PGPR strains to enhance plant growth under drought stress conditions. Such studies emphasize the necessity of comprehending the mechanisms behind demonstrated PGPR-mediated drought tolerance via maintaining the high relative water content.

### 2.3. Improvement of Photosynthesis Rate

The plant’s primary response to drought stress is the promotion of stomatal closure to reduce water loss by transpiration [35]. The negative effect of early partial stomatal closure is the reduction in CO_2_ assimilation, which leads to a decrease in net photosynthetic capacity, but the favorable effect is a drop in transpiration rate, which helps to preserve water in plants [36]. During drought stress, the rate of assimilatory surface growth slows first, followed by photosynthesis inhibition in plants. Drought-induced reduction in plant photosynthetic rate can be attributable to stomatal and non-stomatal limitations. Stomatal photosynthetic limitations are inadequate CO_2_ assimilation rate in plant leaves and sub-stomatal cavities caused by early partial stomatal closure. Non-stomatal photosynthetic limitations occur when there is a decline in chloroplast activity, leaf nitrogen, synthesis of adenosine triphosphate (ATP), ribulose1,5-bisphosphate carboxylase/oxygenase (RuBisCO) activity, ribulose 1,5-bisphosphate (RuBP) synthesis, and impairment of photosystem I and II reactions. However, the effects of non-stomatal photosynthetic limitations are prominent during severe drought stress and the imposed limitations are difficult to alleviate and require prolonged recovery time.

The drought stress restricts CO_2_ uptake, which results in a decline in photosynthetic rate and excess ROS accumulation in various organelles found in plants, especially chloroplasts, peroxisomes, and mitochondria [37]. Excessive ROS accumulation disrupted the structure of thylakoid membranes, enzyme activity, and photosynthetic pigments. Under drought stress, PGPR increases the stomatal conductance, photosynthetic rate, and maximum potential quantum efficiency of photosystem II (Fv/Fm), and decreases the transpiration rate in plants by producing the phytohormones, which affect the essential structural and functional characteristics of their photosynthetic apparatus [33,38,39]. Liu et al. [40] demonstrated the 11.6% enhancement in stomatal conductance, 13% elevation of photosynthetic rate, and the reversal process of chlorophyll degradation when the drought-stressed *Sambucus williamsii* plants were inoculated with cytokinin producing *Acinetobacter calcoaceticus* strain. Thus, it may be deduced that PGPR help to improve the photosynthetic rates under drought stress due to their ability to produce phytohormones.

### 2.4. Production of Phytohormones

The phytohormones, such as auxin, cytokinin, gibberellin, abscisic acid (ABA), ethylene, salicylic acid, and jasmonic acid, are produced regularly in the plant system for their overall growth and development in response to their environmental stresses. The plant growth is promoted by auxin, cytokinin, and gibberellin, but ABA and ethylene inhibit growth [41]. The drought stress reduces the level of these phytohormones, which has several negative impacts on plants [29]. However, the application of PGPR can facilitate regulating the level of phytohormones by synthesizing and secreting them to improve the root system architecture of plants and assist them to thrive during drought (Figure 2) [42,43]. The reduction in ethylene production, altering the normal balance of cytokinins and ABA, and modification of the movement and concentration of IAA in host plants are associated with the endurance to drought stress upon applying PGPR. These changes may facilitate the further investigation of PGPR-mediated drought tolerance in plants. Moreover, the bacterially produced growth regulating phytohormones contributed to host plant metabolism and respiration [44,45]. The different PGPR strains for modulating the plant hormonal levels in mitigating drought stress are listed in Table 1.

#### 2.4.1. Auxin

Drought stress drastically reduced auxin accumulation in plant tissues, which may benefit plants under stress. The diverse bacterial species synthesize auxin phytohormone as indole-3-acetic acid (IAA) in the presence of its suitable precursor L-tryptophan excreted from the plants [53]. The IAA is known to modify the root system architecture of host plants by enhancing the root length, total root surface area, and a number of root tips, which lead to improving the absorption of water and nutrients, as well as coordinating the cellular defense against drought stress [26]. It is estimated that about 80% of PGPR can have the ability to synthesize IAA as a secondary metabolite through a tryptophan-dependent IAA biosynthetic pathway. The bacterial production of IAA alters root system architecture by enhancing the number of root tips and surface area, thus improving root water and nutrient acquisition which assist the plants to better cope with severe drought stress [26,47]. Therefore, it was well documented that the bacterial IAA may directly affect combating drought stress in plants and other phytohormones.

#### 2.4.2. Cytokinin

Cytokinin is a phytohormone that can promote the stomatal opening, decrease root growth, and stimulate shoot growth. The drought stress significantly reduces the natural cytokinin content of plants in association with stomatal closure [54]. Reduced cytokinin content facilitates the possible plant adaptive responses by decreasing water loss and boosting root growth, encouraging soil water exploration. The plants treated with cytokinin producing PGPR have demonstrated their potential importance in relieving the inhibitory effects of drought stress [48,49]. Under drought conditions, increased cytokinin content in PGPR treated plants may contribute to stronger stomatal opening promotion or stomatal closure inhibition, thereby improving photosynthetic capacity and facilitating plant growth [40]. Hence, it was suggested that the cytokinin producing ability of rhizobacteria was believed to be a major plant growth affecting factor correlated with drought stress resistance.

#### 2.4.3. Gibberellin

Gibberellin is a plant hormone that acts as the key plant growth regulator and promotes fruit ripening and seed germination. In addition to promoting plant growth, the hormone acts as a stress protectant. It can scavenge ROS and assist the plants with their greater negative water potential during drought stress, sustaining photosystem II photochemical efficiency [55]. Like other hormones, gibberellins producing PGPR can regulate the endogenous gibberellin content in host plants [51,56,57]. These strains compensate for the lack of plant gibberellin by incorporating bacterial gibberellin. Despite the existence of a few PGPR strains that secrete gibberellin, their significance in mitigating drought effects in host plants remains unknown.

#### 2.4.4. Abscisic Acid

Abscisic acid (ABA) is a stress hormone that accumulates prodigiously in response to drought stress to regulate plant water balance and cellular dehydration tolerance via induction of genes and stomatal closure [35]. The ABA is rapidly produced in the chloroplasts and roots of the plant in response to drought stress. It cannot cross the plasma membrane but can be transferred into stomatal guard cells and promote stomatal closure. The enhancement of ABA contents in stomatal guard cells could lead to a decline in water loss under drought situations. The ABA-mediated stomatal closure reduces CO_2_ uptake, consequently decreasing the photosynthetic rate [58]. Moreover, ABA enables the plants to cope better with drought stress by coordinating with other plant hormones [59]. The PGPR inoculation enhances drought stress tolerance by enhancing the ABA accumulation in plants, which further leads to a lower leaf transpiration rate via stomatal closure [35,52]. Therefore, it may suggest that ABA and/or its analog producing rhizobacteria can modulate the phytohormonal status, stimulate growth, and govern drought stress responses in plants.

#### 2.4.5. Salicylic Acid

Salicylic acid is a versatile stress responsive plant hormone that is significantly produced in plants to elicit the IST against biotic and abiotic stresses [60,61]. It is intriguing to assume that bacterial salicylic acid production may play a defensive role in abiotic stress (particularly drought) resistance by contributing to the endogenously produced plant salicylic acid pool and its signaling pathway [26,62]. Although many PGPR produces salicylic acid and induces tolerance to both biotic and abiotic stresses, there is limited evidence that the bacterially produced salicylic acid plays a direct role in drought tolerance.

#### 2.4.6. Jasmonic Acid

Jasmonic acid is another key plant hormone that improves the plant drought stress tolerance owing to its significant role in regulating stomatal closure. The rapid increase in endogenous jasmonic acid content is the immediate plant response to drought stress, but the content decreases to its basal level with stress prolongation [63]. The bacterial strains isolated from the roots of *Helianthus annuus* produced jasmonic acid under drought conditions, further implying that this ability of bacterial strains may have contributed to significant plant growth promotion and stress tolerance in water stress conditions by enhancing jasmonic acid production [30,64].

### 2.5. Production of Exopolysaccharides

Exopolysaccharides are natural polysaccharides secreted by bacteria, enabling them to thrive in a water scarce environment by holding a huge amount of water [65,66]. They establish the attachment of bacteria to plant root systems, soil particles, and other bacteria. The bacterial secreted exopolysaccharides help to avoid desiccation in drought-stressed plants by producing hydrophilic biofilms on their root surface, which may act as an additional layer to protect the roots and, consequently, allowing for a slower rate of water release in drought stress [67,68]. The exopolysaccharide-producing rhizobacteria could relieve the adverse effects of drought stress in plants due to the better aggregation of rhizosphere soil in the environment, thereby enhancing the plant uptake of water and nutrients for their growth and survival in adverse stress conditions (Figure 3) [36,69,70]. Many studies inferred that the exopolysaccharides producing rhizobacteria could induce drought stress tolerance by their better root colonization and increasing relative water content, proteins, and sugars in inoculated plants to combat osmotic and oxidative stresses under drought stress [33,71,72,73]. Ansari et al. [68] have revealed that the inoculation of *T. aestivum* plants with exopolysaccharides (biofilm) producing *P. azotoformans* contributed to producing ACC deaminase and IAA, enhancing the root colonization, plant growth attributes, photosynthetic pigment efficiency and physiological attributes, and reduce the activity of antioxidant enzymes for drought stress alleviation. Therefore, it can be demonstrated that exopolysaccharides producing PGPR play a most imperative role in drought tolerance, consequently improving global food security. The exopolysaccharide producing PGPR strains to alleviate plant drought stress are summarized (Table 2).

### 2.6. ACC Deaminase Activity for Reducing Ethylene Levels

Ethylene is a key plant stress hormone produced in response to drought stress and is involved in fruit ripening, stress signaling, and sex determination in plants [74]. However, this hormone is detrimental to plant growth and development when present in large amounts, especially in drought. The negative effects of ethylene are leaf abscission induction, stomatal conductance decline, reduced root and shoot growth, epinasty, prolonged root hypoxia, cell damage, and eventual plant death. The bacteria alleviate the negative effects of stress induced ethylene by sequestering and degrading the plant ethylene precursor, 1-aminocyclopropane-1-carboxylic acid (ACC), before its oxidation by ACC oxidase, thereby improving plant growth promotion and stress tolerance (Figure 4) [75,76]. The bacterial ACC deaminase enzyme degrades ACC into α-ketobutyrate and ammonia, thus, imperatively regulating the endogenous ethylene level below the inhibitory level [12,77,78]. The inoculation with ACC deaminase producing PGPR altered the plant biomass, relative water content, photosynthetic rate, phytohormones contents, ACC content, ethylene emission, membrane lipid peroxidation, production of ROS scavenging enzymes, and osmoprotectantsfor enabling the plants’ ability to withstand drought stress [36,79,80]. Murali et al. [81] have demonstrated that ACC deaminase producing *B. amyloliquefaciens* could protect *P. glaucum* plants against severe drought conditions via inducing an antioxidant defense system, thus promising its application in the management of drought stress. Therefore, it has been shown that PGPR use ACC deaminase production as a key mechanism for alleviating drought stress in plants and ensuring their survival, as well as overall growth and development. The ACC deaminase producing PGPR strains for mitigating drought stress in plants are listed (Table 2).

### 2.7. Production of Carotenoids

Carotenoids are the major class of secondary metabolites produced in plants that are essential for survival in drought stress conditions [93,94]. They serve as antioxidants against different types of ROS generated and help to activate the plant antioxidant defense system. The carotenoids also strengthen the thylakoid rigidity and cell membrane by acting as antioxidants [95]. Moreover, they serve as precursors for the biogenesis of two phytohormones, such as ABA and strigolactones [96]. The plant species, duration, and intensity of drought stress substantially influence carotenoids production and regulation. Yasmin et al. [97] have reported that the carotenoids were significantly enhanced in *Z. mays* plants treated with the PGPR strains *B. pumilus* and *Pseudomonas* sp. under both drought-stressed and non-stressed conditions. Batool et al. [98] have suggested that *S. tuberosum* plants treated with PGPR *B. subtilis* were found to show a higher content of carotenoids than those without PGPR under drought conditions. Specifically, the production of carotenoids is critical for the plant’s photosynthetic machinery, and drought stress greatly influences the content; however, the carotenoid content in plants can be enhanced with the PGPR treatment.

### 2.8. Regulation of Emission of Volatiles

Volatiles are commonly emitted from the plant leaves as a stress tolerance response when subjected to various stresses [99]. The stress-induced volatiles act as external signals to elicit direct and indirect defensive responses within the same and surrounding plants. The drought stress can cause stomatal closure, which reduces constitutive volatile emissions and activates the plant defense systems. The emission rates of a bouquet of constitutive volatiles in plants are often substantially related to the severity of drought stress [100]. Thus, monitoring stress-induced emission of plant volatiles might provide key insight into their immediate stress responses, priming and acclimatization to the stress. The volatile bacterial *2R,3R*-butanediol produced by *P. chlororaphis* was considered the major determinant in eliciting drought resistance in *A. thaliana* plants via salicylic acid dependent mechanism [101]. The inoculation of *T. aestivum* with *B. thuringiensis* resulted in significantly greater survival of drought-stressed plants and increased photosynthesis and biomass production due to significantly reduced emission of stress-induced volatiles [38]. Brunetti et al. [90] have implied that the beneficial effect of PGPR inoculation on the growth of *M. pruriens* was also evident from promoting the isoprene emission, thus augmenting the plant tolerance to water stress. The bacterial volatiles is particularly promising candidates as priming agents for a rapid, non-invasive approach to evaluating the plant drought stress severity and its alleviation in the early stages of stress development [102]. Hence, it was suggested that rhizobacterial inoculation improved plant drought stress tolerance by reducing stress-induced volatiles emission.

### 2.9. Enhancement of Uptake and Assimilation of Mineral Nutrients

Drought stress adversely affects the uptake and assimilation of mineral nutrients, reducing their contents in plant tissues and eventually causing impaired plant growth and development [103,104]. The deficiency of mineral nutrients (macronutrients and micronutrients) directly correlates with decreased plant yields in drought conditions as they are passed by water movement to the roots. The water-soluble mineral nutrients (such as potassium, phosphorus, nitrogen, sulfur, magnesium, calcium, silicon, and zinc) were reported to alleviate the negative effects of drought stress by mediating many processes, such as photosynthesis, osmoregulation, protein synthesis, and enzyme activation [105,106]. The inoculation of native PGPR strains enhances the growth of autochthonous shrubs strongly associated with nutrient acquisition from the soil without altering microbial diversity in the rhizosphere [107,108,109]. Moreover, the PGPR properly supplements the mineral nutrients to crop plants that can help to minimize drought stress by actively participating in defensive processes, such as antioxidant systems and osmoregulation [33,90,110]. Therefore, the evidence suggests that PGPR-mediated uptake and enrichment of essential nutrients in plants play a pivotal role in enhancing their growth and drought resistance.

### 2.10. Siderophore Production

The siderophores are low-molecular weight secondary metabolites with the capacity to chelate iron. The siderophore produced by PGPR assists in the fulfillment of iron requirements of plants by solubilizing and chelating iron from organic or inorganic complexes present in soil [17,20]. The siderophore-producing PGPR protects the plant from phytopathogens at the rhizosphere through iron starvation and competitive exclusion under iron-deficient conditions [21,28]. Moreover, they can also detoxify the heavy metal contamination by mobilizing insoluble heavy metals, viz., aluminum, copper, cadmium, chromium, lead, and mercury [14,15]. In addition, Ferreira et al. [16] showed that PGPR exert plant growth promoting activity by producing extracellular siderophores that deprive the pathogens of iron nutrition. The studies indicate that PGPR producing siderophores function as biofertilizers or biocontrol agents that are environmentally friendly, thereby rendering their hand in sustainable agricultural crop production.

### 2.11. Induction of Antioxidant Defense Systems

The higher production of ROS in different cellular compartments is the inevitable effect of the drought stress response of plants [111]. The most well-known ROS, such as hydroxyl radical (OH^•^), superoxide radical (O_2_^•−^), singlet oxygen (^1^O_2_), and hydrogen peroxide (H_2_O_2_), accumulate as a result of partial reduction in atmospheric oxygen (O_2_) within the photosynthetic electron transport chain in both mitochondria and chloroplasts. The ROS produced can cause progressive oxidative damage to proteins, nucleic acids, and lipids, ultimately resulting in plant cell death. It is well known that ROS can have dual roles in plants, as they act as secondary messengers under stress conditions at their lower concentrations, but they are toxic by-products and progressively cause oxidative damage at their higher concentrations [1]. The plants possess enough antioxidant enzymes and non-enzymatic antioxidants, reducing excess ROS accumulation in different organelles and lowering oxidative damage under drought conditions [29]. The ROS scavenging antioxidant enzymes (such as ascorbate peroxidase (APX), glutathione reductase (GR), dehydroascorbate reductase (DHAR), catalase (CAT), monodehydroascorbate reductase (MDHAR), peroxidase (POX), and superoxide dismutase (SOD)) alleviate the negative consequences of excess ROS accumulation by oxidation, reduction and dismutation of O_2_ to H_2_O_2_ [112,113]. Therefore, the drought stress alters and reduces the activity of these ROS scavenging antioxidant enzymes, rendering them incapable of performing ROS regulating functions.

The PGPR notably increase the activity of ROS scavenging antioxidant enzymes, hence decreasing the excess ROS accumulation in drought stress affected plants [114]. It was proposed that the plants treated with PGPR could endure drought stress due to the enhancement of the activity of ROS scavenging antioxidant enzymes. Batool et al. [98] have revealed that PGPR *B. subtilis* was found to improve the activity of antioxidant enzymes CAT, SOD, and POD by 68%, 63%, and 51%, respectively, in severe drought stressed *S. tuberosum* plants as compared to the plants without PGPR treatment. Hence, it can be easily recognized that PGPR can reduce oxidative stress in drought stress affected plants by interfering with the regulation of the antioxidant defense system. Therefore, enhanced activity of ROS scavenging antioxidant enzymes might be the primary cause of drought tolerance in PGPR treated plants.

In addition, the major low molecular weight antioxidants, such as alkaloids, ascorbic acid (vitamin C), carotenoids, flavonoids, glutathione, phenolic compounds, non-protein amino acids, and α-tocopherol (vitamin E), are produced in plant tissues for ROS scavenging and protected themselves against drought stress [115]. Ghorbanpour et al. [116] have revealed that the inoculation of *Hyoscyamus niger* plants with rhizobacterial strains (*P. fluorescens* and *P. putida*) remarkably improved the production of tropane alkaloid in plants under water deficit stress, resulting in contributing to their stress resistance. It was noted that the total phenolic content in PGPR treated drought stressed plants was higher than untreated stressed plants, thereby providing evidence that the antioxidant system plays a vital role in drought tolerance [117]. The PGPR-mediated induced antioxidant defense system for drought stress tolerance in plants is represented in Figure 5.

### 2.12. Reduction in Lipid Peroxidation and Electrolyte Leakage

Excess ROS production causes increased membrane lipid peroxidation and consequent damage to proteins, nucleic acids, and lipids in drought conditions [111]. The rate of lipid peroxidation is increased in plants by increasing malondialdehyde (MDA, end-product of lipid peroxidation) accumulation owing to oxidative damage to mitochondria and chloroplasts. The MDA accumulation and electrolyte leakage are important to stress signals which considerably increase under drought stress, resulting in detrimental effects on the selective permeability and stability of cell membrane [118]. In this regard, the PGPR application led to decreased MDA accumulation, electrolyte leakage and cell membrane damage, and increased membrane stability due to the enhancement of the activity of antioxidant enzymes and non-enzymatic antioxidants in the drought exposed plants [114,117,119]. Therefore, it has also been stated that the significant reduction in MDA accumulation and electrolyte leakage in PGPR inoculated drought stressed plants are also used to determine their stress tolerance.

### 2.13. Osmotic Adjustment

Plant cells use an osmotic adjustment as an adaptive mechanism by maintaining cell turgidity to boost their resistance to drought stress [120]. The most important mechanisms to maintain osmotic balance are osmoregulation and osmoprotection. Osmoregulation is the process of maintaining cell turgor by lowering osmotic potential (Ψ) via synthesis and accumulation of osmoprotectants (or osmolytes), such as soluble proteins, amino acids, quaternary ammonium compounds, polyamines, sugars, and sugar alcohols in plants [121,122]. Osmoprotection either boosts the antioxidant defense system or assists with ion homeostasis. The better cellular ion homeostasis involves the enhancement of biosynthesis and accumulation of different osmoprotectants (such as proline, trehalose, glycine betaine, sugars, and polyamines) in the plant cytoplasm to maintain the cell turgidity and normal cellular functions [123,124]. The drought stress interferes with the osmotic adjustment mechanism by affecting the plant’s inherent ability to produce these osmoprotectants. As a result of drought stress, PGPR secretes many osmoprotectants (such as amino acids, proline, and soluble sugars), which synergistically function with plant-produced osmoprotectants to boost plant growth [71,98,114]. The PGPR could enhance the accumulation of osmoprotectants in plants through de novo biosynthesis or direct environmental acquisition to help them against drought stress with the mechanism of osmotic adjustment.

Proline is a major osmoprotectant that accumulates in plant cells during drought stress to osmotic adjustment and to protect them from oxidative stress via ROS scavenging [125]. It acts as a cytosolic pH buffer to maintain the cell turgor and stabilize the sub-cellular structures (such as proteins, enzymes, and lipids) to counteract the negative effects of drought in plants. The drought stress also reduces nitric acid absorption and transportation to leaves, resulting in lower proline synthesis [126]. In combination with sugars, proline protects the plant photosystems I and II against excessive oxidation during drought [127]. The enhancement of osmotic adjustment by increasing proline level in *Streptomyces pactum* inoculated wheat plants indicated improved plant tolerance towards drought stress [128]. The PGPR-mediated proline accumulation indicates the plant’s key mechanism to better cope with drought stress [129].

Trehalose is a highly stable non-reducing disaccharide that acts as an osmoprotectant to improve the plant’s drought tolerance. It can impart osmoprotection and help to mitigate the drought impacts by preventing protein aggregation and degradation in plants that occur when they are subjected to stress [118]. It has also been demonstrated to protect plant photosystem II from excessive oxidation and regulate ABA metabolism in drought conditions [130]. The drought tolerance and significant increase in plant biomass are conferred by inoculating *Z. mays* plants with *Azospirillum brasilense* over-expressing trehalose biosynthetic gene [131]. Thus, it was well established that the bacterial trehalose could play the main role as a signaling molecule to improve plant drought tolerance.

Choline is a precursor of glycine betaine which essentially promotes the production and accumulation of glycine betaine for plant stress resistance. Glycine betaine is a potent osmoprotectant that is biosynthesized from choline through a two-step oxidation reaction and plays a crucial role in plant stress tolerance by maintaining intermolecular water balance and stabilizing cellular macromolecules [132]. The accumulation of glycine betaine (like the quaternary ammonium compound) increases the osmotic adjustment for drought stress adaptability of plants and is highly dependent on the amount of choline available in chloroplasts. The substantial effect of glycine betaine on lowering the ROS level (such as O_2_^•−^ and H_2_O_2_) MDA and electrolyte leakage was demonstrated to protect the plasma membrane [133]. The beneficial soil bacterium *B. subtilis* increases plant choline and glycine betaine synthesis via induced gene expression levels of key enzymes, such as phosphoethanolamine *N*-methyltransferase (PEAMT), associated with augmented plant tolerance to osmotic stress caused by dehydrating conditions [134]. The PGPR has promoted the accumulation of glycine betaine that regulates the plant stress responses by reducing water loss caused by osmotic stress during drought [135]. Glycine betaine does not directly scavenge ROS; instead, it produces H_2_O_2_, stimulating ROS scavenging antioxidant enzymes, thereby reducing oxidative stress.

The accumulation of soluble sugars as osmoprotectants is another drought adaptive mechanism that contributes to osmotic adjustment in plants [127]. It was well known that starch hydrolysis leads to an increase in the soluble sugar content in drought stress affected plants. The plants treated with PGPR strains displayed the accrual of soluble sugars due to starch degradation, which resulted in drought resistance because the soluble sugars acted as an osmoprotectant under drought conditions [114,136]. Similarly, the natural polyamines (such as spermidine, spermine, and putrescine) are low molecular weight aliphatic amine compounds that are ubiquitously considered to be the plant growth regulating and protective compounds involved in defense response to abiotic stresses, especially drought [126]. Zhou et al. [137] have revealed that the spermidine producing rhizobacterial strain *B. megaterium* was displayed to enhance the growth and drought resistance correlated with modulation of cellular ABA levels and activation of adaptive responses of host plants. In addition to other polyamines, cadaverine producing *A. brasilense* strain was also involved in the plant growth promotion and osmotic stress alleviation in hydroponics conditions [138]. The bacterial production of polyamines correlated with plant growth promotion and osmotic stress alleviation.

Consequently, it can be perceived that the PGPR aid in synthesizing and accumulating such osmoprotectants, allowing the plants to enable osmotic adjustment and alleviate drought stress. Thus, the PGPR secreted excess osmoprotectants have drawn the attention of researchers worldwide as the plant survival mechanism under harsh drought conditions.

### 2.14. Alteration in Expression of Stress Responsive Genes

The strong induction of stress responsive gene expression is critical for plant survival during drought [139,140]. Recently, molecular approaches have been employed to study the gene expression and physiological functions involved in plant tolerance induction against drought stress [78,81]. Using quantitative real-time PCR (qRT-PCR), the stress-related gene upregulation in bacterial colonized plant leaves is detected as the increased activity of enzymes provides drought stress tolerance when the plants are primed with PGPR, thereby alleviating the deleterious effects of drought stress. The qRT-PCR analysis showed that PGPR inoculation confers greater drought tolerance by positively modulating differential gene expression involved in ethylene biosynthesis (*ACS* and *ACO*), jasmonate (*MYC2*) and salicylic acid (*PR1*) signaling, ROS scavenging (*APX*, *CAT*,and *GST*), stress response (*DHN* and *LEA*), and transcription activation (*DREB1A* and *NAC1*) in plants exposed to drought stress [79,139,141]. Li et al. [128] have revealed that *S. pactum* increased the leaf ABA content and upregulated the drought stress resistance gene expression (such as *EXPA2*, *EXPA6*, *P5CS*, and *SnRK2*), thereby reducing the effects of drought stress in *T. aestivum* plants.

Further, Omar et al. [142] have shown that the inoculation of *Oryza sativa* plants with *B. megaterium, P. azotoformans* and *Rhizobium* sp. was found to enhance drought stress tolerance via altering the expression of plant growth and stress-related genes (such as *AP2-EREBP*, *COX1*, *DHN*, *EXP1*, *EXP2*, *EXP3*, *GRAM*, *GST*, *NAM*, and *NRAMP6*). Transcriptomic analysis was achieved to assess the beneficial effects of PGPR inoculation on the entire set of genes and to identify plant affected pathways [141] generally. Moreover, the inoculation of PGPR mainly influenced the dependence of transcriptomic profile on the plant genotypes, bacterial strains, and drought levels. It could be concluded that the bacterial inoculation activated the expression of stress-related genes to confer greater drought resistance in plants. The several PGPR strains that altered the expression of stress-related genes for the induction of drought tolerance in plants are represented (Table 3).

## 3. Future Prospects

In the current global climate change scenario, exploiting beneficial plant–microbe interactions is an essential strategy to improve food production to feed the world’s rising population. With improved crop production, soil fertility, and sustainable agriculture, researchers are shifting towards rhizosphere engineering to develop a unique environment for plant–microbe interactions. The rhizosphere engineering using PGPR offers various applications, including crop fertilization and developing environmentally friendly sustainable agriculture practices [144,145]. The literature suggests that PGPR is well known to mitigate abiotic stresses (especially drought) and improve total plant growth and health [146,147,148]. Future research should focus on developing effective microbial formulations with a longer shelf life to achieve sustainable agricultural practices in drylands that extensively minimize the usage of chemical fertilizers and pesticides. However, the constancy of PGPR performance is still questionable in the dry field conditions due to water scarcity. The sources of this problem are the low-quality inocula and bacterial inability to compete with indigenous populations in extreme environmental stress conditions. The inoculated beneficial PGPR strains must be particularly adept at rhizosphere to compete with native populations to procure scarce resources, survive, and colonize the specific root regions.

The current research interest should be focused on isolating native PGPR strains from the stress affected soils that might be exploited as biofertilizers and bioinoculants for agricultural crop production in dry lands. Moreover, the inoculation method is particularly significant because improper application can result in inconsistent and false results. The use of seed coating, liquid inoculation, and peat-based inoculants are some of the accurate inoculation techniques. The selection and performance of robust bacterial strains in relatively changeable climatic conditions are essential for the successful commercialization of PGPR which depends on their shelf life, cell viability, protection in an adverse environment, cost-effectiveness, and convenient use, as well as their non-pathogenic and non-toxigenic nature. Overall, the research has been directed towards the PGPR application in drought stress affected areas and encourages the large-scale commercialization of bioinoculants in sustainable agriculture practices. Despite recent advances and perspectives on PGPR-mediated plant drought stress tolerance, we are still beginning to understand their mechanisms conferring greater tolerance. However, the current research indicates that future study has a high potential for providing new insights into the environmental sustainability of food production.

## 4. Conclusions

Because drought stress has such detrimental impacts on plant growth and productivity, many drought-monitoring techniques have been developed, each with its advantages and disadvantages. In addition to offering additional direct and indirect benefits to improve plant performance significantly, the PGPR application is most successful in overcoming drought. A better understanding of the mechanisms of PGPR-mediated drought stress responses in plants is crucial for enhancing agricultural crop yield, as well as solving future global food security issues. However, the commercial application of PGPR greatly depends on its cost-effectiveness, robustness property of strain, and shelf life of formulation product. Despite that, the promise of PGPR’s beneficial effects on drought mitigation in plants cannot be ignored. Therefore, greater attention must be given to making efficient PGPR-based formulations commercially available to typical farmers worldwide to improve dry-land agriculture.

## Figures and Tables

**Figure 1 antioxidants-11-01763-f001:**
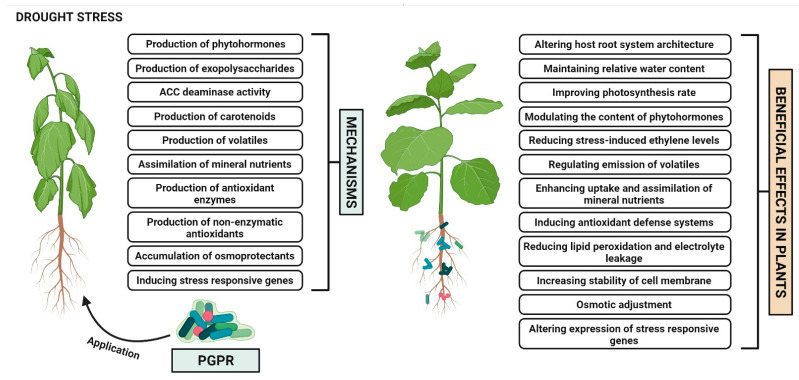
Mechanisms of PGPR-mediated induction of plant drought stress tolerance.

**Figure 2 antioxidants-11-01763-f002:**
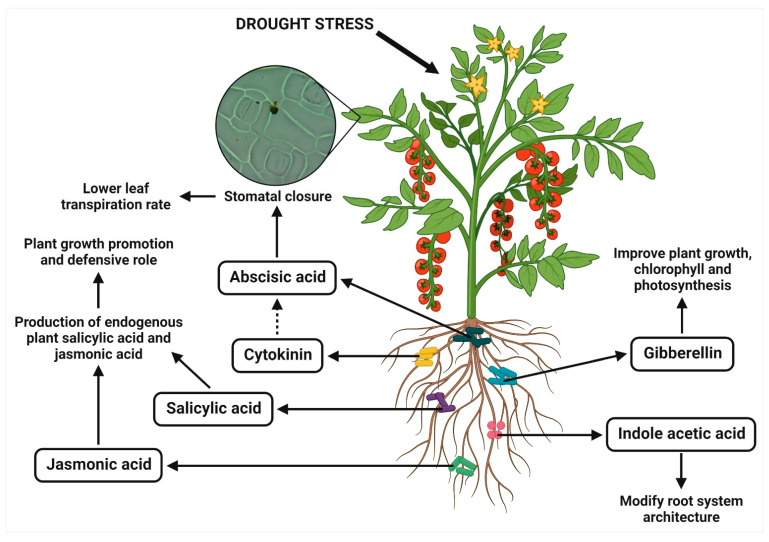
PGPR-mediated modulation of phytohormones for enhancing drought tolerance in plants.

**Figure 3 antioxidants-11-01763-f003:**
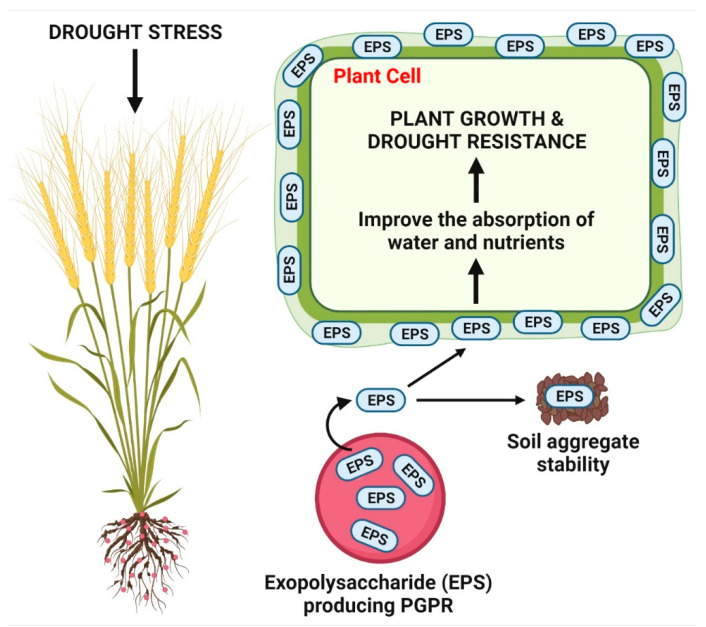
Exopolysaccharide producing PGPR-mediated alleviation of drought stress in plants.

**Figure 4 antioxidants-11-01763-f004:**
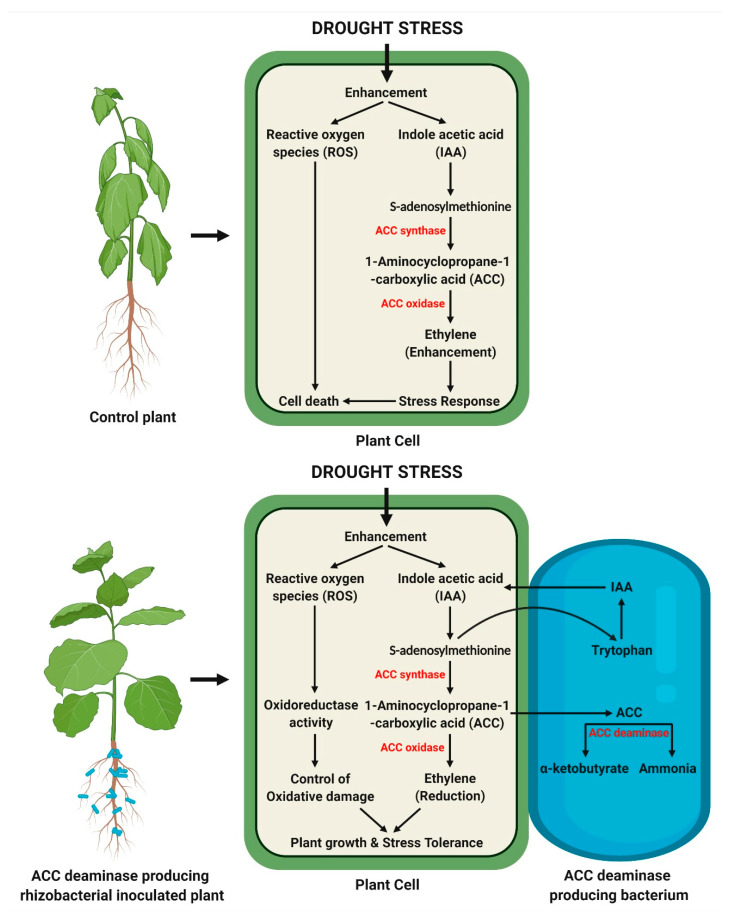
ACC deaminase producing PGPR-mediated mitigation of drought stress in plants.

**Figure 5 antioxidants-11-01763-f005:**
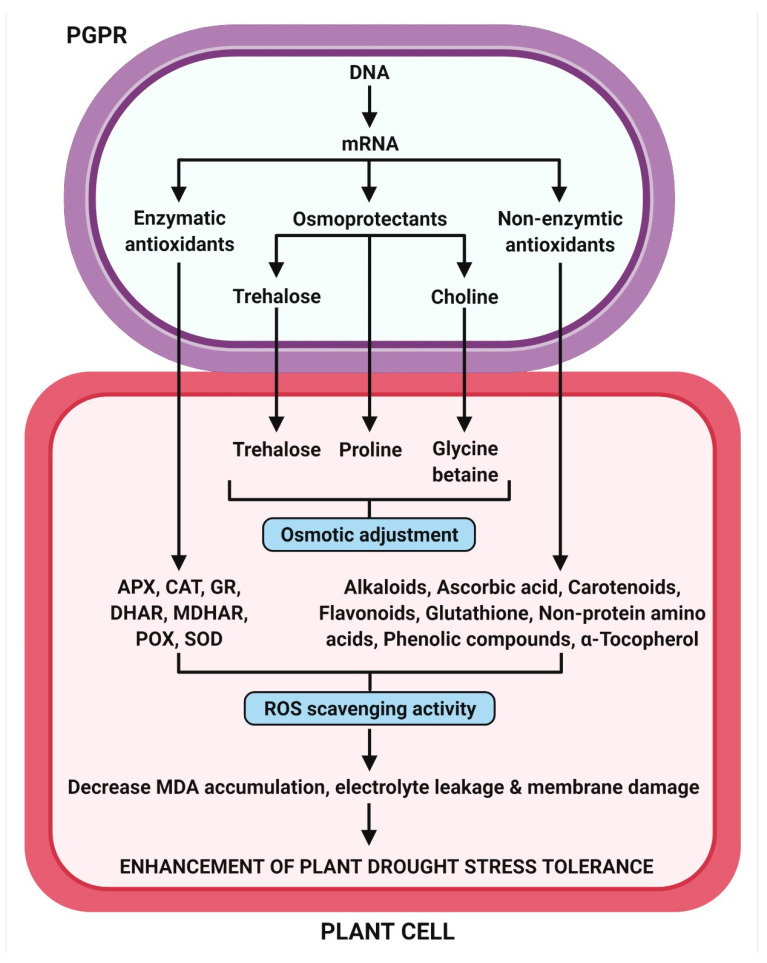
PGPR-mediated induced antioxidant defense system for the drought stress tolerance in plants.

**Table 1 antioxidants-11-01763-t001:** PGPR-mediated production of phytohormones in enhancing plant drought stress tolerance.

PGPR Strains	Plant Species	Mechanisms	Beneficial Effects	Reference
*Bacillus megaterium* and *Pseudomonas putida*	*Trifolium repens*	IAA production	Enhanced the water content, root and shoot biomass	[24]
*Bacillus amyloliquefaciens*, *B. muralis*, *B. pumilus*, *B. simplex*, *B. thuringiensis*, *Enterobacter aerogenes*, *Moraxella pluranimalium*, and *P. stutzeri*	*T. aestivum*	IAA production	Significantly improved the shoot length, spike length, seed weight, tillers and number of spikelets, and increased the peroxidase, acid phosphatase and proline content in plants	[46]
*Bacillus* sp. and *Enterobacter* sp.	*T. aestivum* and *Z. mays*	IAA and salicylic acid production	Displayed the root system architecture alteration *viz.,* increased the number of root tips, root surface area, root length and root branching	[26]
*P. aeruginosa*	*Vigna radiata*	IAA production	Enhanced the shoot length, number of grains, pod/plant, total yield, 100 seed weight and 100 seed straw weight, improved the photosynthetic activity, membrane stability, relative water content and antioxidant efficacy	[47]
*B. subtilis*	*Lactuca sativa*	Cytokinin production	Increased the shoot cytokinins, stimulated the shoot mass accumulation and shortened roots, decreased stomatal conductance and root/shoot ratios	[48]
*B. subtilis*	*Platycladus orientalis*	Cytokinin production	Showed the higher leaf relative water content and water potential, increased the plant root exudates (like amino acids, sugars and organic acids) and stomatal conductance, and elevated the levels of cytokinins in shoot	[49]
*A. calcoaceticus*	*S. williamsii*	Cytokinin production	Increased the photosynthetic rate, decreased the stomatal conductance and intracellular CO_2_ concentration	[50]
*P. putida*	*Glycine max*	Gibberellin production	Increased the shoot length and fresh weight, higher chlorophyll content, lower levels of ABA and salicylic acid, higher jasmonic acid level, reduced sodium content, increased phosphate content and modulated the antioxidants by decreasing radical scavenging activity, SOD and flavonoids	[51]
*B. licheniformis* and *P. fluorescens*	*Vitis vinifera*	ABA production	Increased the plant ABA levels, diminished the water loss rate and incremented the synthesis of defense-related terpenes	[52]
*B. marisflavi*	*Brassica juncea*	ABA analogue/xanthoxin production	Delayed the drooping points of plants and higher drought stress tolerance index, induced the stomatal closure, inhibited the seed germination, and decreased the gibberellic acid induced α-amylase activity	[35]

**Table 2 antioxidants-11-01763-t002:** ACC deaminase and exopolysaccharide producing PGPR in the alleviation of plant drought stress.

PGPR Strains	Plant Species	Mechanisms	Beneficial Effects	Reference
*B. subtilis*	*Trigonella foenum-graecum*	ACC deaminase	Improved the plant growth and nutrient uptake, reduced the ACC level, alleviated the ethylene induced damage, increased the rhizobial nodulation and arbuscular mycorrhizal fungal colonization in plants	[82]
*Achromobacter xylosoxidans*, *P. oryzihabitans* and *V. paradoxus*	*Solanum tuberosum*	ACC deaminase	Increased the plant root biomass and tuber yield, decreased the concentrations of rhizosphere ACC and proteinogenic amino acids exuded from the plant root	[83]
*P. putida*	*Z. mays*	ACC deaminase	Enhanced the growth of seedlings, root colonization and improved the stomatal conductance and cellular metabolites	[84]
*Burkholderia* sp. and *Mitsuaria* sp.	*Arabidopsis thaliana* and *Z. mays*	ACC deaminase and exopolysaccharide production	Altered the root structure system, reduced the evapotranspiration, and modified the level of proline, MDA, phytohormones, and activity of antioxidant enzymes	[25]
*P. fluorescens* and *P. palleroniana*	*T. aestivum*	ACC deaminase	Enhanced the plant growth, foliar nutrient content, chlorophyll content, proline content and activity of antioxidant enzymes (APX, CAT, GPX, and SOD), and decreased MDA and H_2_O_2_ content	[85]
*Enterobacter hormaechei*, *P. fluorescens*, and *P. migulae*	*Setaria italica*	ACC deaminase and exopolysaccharides production	Stimulated the seed germination and seedling growth, efficiently colonized the root adhering soil and enhanced the root adhering soil/ root tissue ratio and soil moisture content	[70]
Consortia of *B. subtilis*, *Ochrobactrum pseudogrignonense*, and *Pseudomonas* sp.	*Pisum sativum* and *Vigna mungo*	ACC deaminase	Significantly enhanced the seed germination, dry weight, shoot length and root length, higher leaf chlorophyll content, improved the root recovery intension and relative water content, decreased the ACC accumulation and elevated the production of ROS scavenging antioxidant enzymes and cellular osmolytes	[79]
*Bacillus* sp. and *Enterobacter* sp.	*Mucuna pruriens*	ACC deaminase	Significantly improved the root and shoot lengths, reduced the ACC content and ethylene emission, and higher emission of isoprene	[12]
*Ochrobactrum anthropi*, *P. fluorescens*, *P. palleroniana* and *Variovorax paradoxus*	*T. aestivum*	ACC deaminase	Increased the shoot growth, root growth, root/ shoot ratio, total chlorophyll content, proline content and total phenolics, exhibited lower H_2_O_2_ content, reduced MDA levels and improved the activity of antioxidant enzymes (APX, CAT, GPX, and SOD) and foliar nutrient contents (calcium, nitrogen, potassium, phosphorus, and sodium)	[80]
*B. subtilis* and *B. thuringiensis*	*H. annuus*	ACC deaminase	Showed the plant growth promotion, increased the proline content and activity of antioxidant enzymes (APX and SOD) and decreased MDA content	[86]
*O. anthropi*, *P. fluorescens*, *P. palleroniana* and *V. paradoxus*	*Eleusine coracana*	ACC deaminase	Improved the overall plant growth parameters and concentration of nutrients, elevated the activity of ROS scavenging antioxidant enzymes (APX, CAT, GPX, and SOD), cellular osmolytes (such as proline and phenol), higher leaf chlorophyll and reduced the level of H_2_O_2_ and MDA	[87]
*B. subtilis*	*Solanum lycopersicum*	ACC deaminase	Improved the plant growth, relative water content, the activity of antioxidant enzymes (APX and SOD) and proline content, and decreased MDA, H_2_O_2_, and superoxide anion accumulation	[78]
*Bacillus velezensis*	*Z. mays*	ACC deaminase and exopolysaccharides production	Significantly showed better root colonization, improved the plant growth and physiological parameters viz., water use efficiency, vapor pressure, stomatal conductance, photosynthesis, and transpiration	[37]
*P. fluorescens*	*Z. mays*	ACC deaminase	Significantly improved the concentration of photosynthetic pigments, Fv/Fm ratio, free proline, total soluble sugars and nutrients uptake, and increased the yield traits	[88]
*P. lini* and *Serratia plymuthica*	*Ziziphus jujuba*	ACC deaminase	Increased the plant height, root and shoot dry matter, relative water content and soil aggregate stability, decreased the levels of MDA and ABA, and increased the activity of antioxidant enzymes (POD and SOD)	[89]
*Bacillus* sp. and *Enterobacter* sp.	*M. pruriens*	ACC deaminase	Showed the higher water use efficiency and total biomass, lower root ACC content and ethylene emission, higher levels of isoprene emission and carbon assimilation	[90]
*Enterobacter soli* and *P. corrugata*	*V. vinifera*	ACC deaminase	Increased the soil aggregate stability, root adhering soil/root tissue ratio, levels of phosphorus and nitrogen in plant leaves and soil, altered the biomass of root and shoot, plant height, relative water content and net photosynthetic rate, and changed the contents of IAA, ABA, and MDA in plants	[36]
*B. amyloliquefaciens*	*Pennisetum glaucum*	ACC deaminase	Increased the seed germination, seedling vigor index, plant growth parameters, total chlorophyll content, relative water content, proline content and activity of antioxidant enzymes (APX and SOD), and decreased the MDA content	[81]
*P. putida*	*H. annuus*	Exopolysaccharides production	Proficiently colonized the root adhering soil and rhizoplane, enhanced the plant biomass, survival rate, root adhering soil/root tissue ratio and percentage of stable soil aggregates	[91]
*B. amyloliquefaciens*, *B. licheniformis*, *B. subtilis*, *B. thuringiensis*, and *Paenibacillus favisporus*	*Z. mays*	Exopolysaccharides production	Increased plant biomass, soil aggregate stability, root adhering soil/root tissue ratio, leaf water potential and relative water content, decreased leaf water loss, enhanced the proline, free amino acids and sugars and decreased the activity of antioxidant enzymes (APX, CAT and GPX) and electrolyte leakage	[71]
*Alcaligenes faecalis*, *Proteus penneri*, and *P. aeruginosa*	*Z. mays*	Exopolysaccharides production	Improved the plant biomass, shoot and root length, soil moisture content, leaf area, relative water content, protein and sugar content, decreased the activity of antioxidant enzymes (CAT, POD and SOD) and proline content	[72]
*B. endophyticus* and *P. aeruginosa*	*A. thaliana*	Exopolysaccharides production	Better root colonization and increased plant water content, fresh and dry weights	[73]
*B. cereus* and *Planomicrobium chinense*	*T. aestivum*	Exopolysaccharides production	Significantly increased the contents of leaf sugar and protein, higher chlorophyll fluorescence (Fv/Fm), chlorophyll content and performance index, and reduced the activity of antioxidant enzymes (APOX, CAT, and POD), proline content and lipid peroxidation, enhanced the relative water content and augmented the accumulation of micro/ macronutrients (such as Ca, Mg, Na, K, Cu, Cr, Zn, and Fe)	[33]
*P. azotoformans*	*T. aestivum*	Exopolysaccharides (biofilm) production	Significantly enhanced the root colonization, plant growth characteristics, physiological attributes and photosynthetic pigment efficiency, and decreased the activity of antioxidant enzymes (CAT, GR, and SOD)	[68]
*P. chlororaphis*	*A. thaliana*	Production of bacterial volatile *2R,3R*-butanediol	Reduced the loss of water via stomatal closure	[92]
*Bacillus thuringiensis*	*T. aestivum*	Reduction in volatile emissions	Enhanced the biomass production, greater photosynthesis and survival of drought stress affected plants	[38]

**Table 3 antioxidants-11-01763-t003:** PGPR-mediated alteration of stress responsive gene expression in plant drought stress tolerance.

PGPR Strains	Plant Species	Mechanisms	Beneficial Effects	Reference
*P. aeruginosa*	*V. radiata*	–	Strongly upregulated the drought stress responsive genes such as catalase (*CAT1*), dehydrin (*DHN*) and dehydration-responsive element binding protein (*DREB2A*)	[143]
*P. putida*	*Cicer arietinum*	Molecular responses	Repressed the expression of transcription activation genes (*DREB1A* and *NAC1*) and stress responsive genes (*DHN* and *LEA*), upregulated the antioxidative enzymes (*APX*, *CAT*, and *GST*), downregulated the ethylene biosynthesis genes (*ACS* and *ACO*), upregulated jasmonate signaling gene (*MYC2*) and salicylic acid signaling gene (*PR1*)	[139]
*B. megaterium* and *Enterobacter* sp.	*S. lycopersicum*	Molecular responses	Reduced the expression of ethylene biosynthesis genes (*ACO*, *ACS*, *ERF*, *ETR*, and *TCTR1*)	[141]
Consortia of *B. subtilis*, *O. pseudogrignonense*, and *Pseudomonas* sp.	*P. sativum* and *V. mungo*	ACC deaminase	Downregulated *ACO* gene expression	[79]
*B. subtilis*	*S. lycopersicum*	ACC deaminase	Significantly decreased the drought responsive gene (*Le25*) and ethylene responsive factor (*SlERF84*)	[78]
*S. pactum*	*T. aestivum*	–	Upregulated the levels of stress resistance gene expression (such as *EXPA2*, *EXPA6*, *P5CS*, and *SnRK2*)	[128]
*B. amyloliquefaciens*	*P. glaucum*	ACC deaminase	Significantly increased the expression of antioxidant enzymes (*APX1* and *SOD1*), and decreased the expression of ethylene-responsive factor (*ERF-1B*) and drought-responsive gene (*DREB-1E*)	[81]
*B. megaterium, P. azotoformans* and *Rhizobium* sp.	*O. sativa*	Molecular responses	Altered the expression of growth and stress related genes (*AP2-EREBP*, *COX1*, *DHN*, *EXP1*, *EXP2*, *EXP3*, *GRAM*, *GST*, *NAM*, and *NRAMP6*)	[142]

Note: *EXPA*—α-expansin, *ACO*—ACC oxidase, *ACS*—ACC synthase, *ERF*—ethylene response factor, *ETR*—ethylene receptor.

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
