# Peer review of "Insight into Recent Progress and Perspectives in Improvement of Antioxidant Machinery upon PGPR Augmentation in Plants under Drought Stress: A Review"

_antioxidants, 2022, doi:10.3390/antiox11091763_

Round 1

Reviewer 1 Report

PGPR have been demonstrated to facilitate plants to resist or tolerant adverse environmental stresses including drought, salt, heavy metal, as well as others in recent years. The application of PGPR to cope with environmental stresses will be a promising strategy in the future. The present manuscript was well written, however, some suggestions are raised as below.

1. The present manuscript have presented multiple mechanisms for PGPR enhancing plants tolerance to the drought stress, so the title of the manuscript should be changed, but not only focusing on the antioxidant mechanism.

2. In the section 2 “Plant growth promoting rhizobacteria mediated drought stress tolerance in plants”, on one hand, I think the introduction for every part involved is very redundant, which should be simplified. On the other hand, siderphores producing PGPR and underlying mechanisms of the interaction with plants of should be elaborated.

3. I think the architecture should be reorganized according to the correlations among multiple mechanisms, because there existed overlapped description.

4. In the line 90, immune responses? I think it should be modified, because there is not any immune response in the following description.

5. Line 128-129, what is relationship between ROS and plants' high relative water content? Is there any published reference?

6. Line 136-139, the relative references should be cited. In addition, the relationship between the production of phytohormones and high water content should be elaborated.

7. There are so many minor errors in the manuscript. If the bacteria genus name showed repeatedly, the name should be abbreviated, and all of the species name should be italic. In addition, line 339 “2R,3R-” should also be italic. The similar errors should be modified through the whole manuscript.

8. What does the sentence in line 406-407 mean?  It should be more precise for convenient reading.

9. There are so many grammar errors in the whole manuscript.

Author Response

Authors’ response to the Reviewer(s) comments

We profusely thank the reviewers for their constructive comments. Herewith we are submitting the revised manuscript following incorporation of all the suggestions and hope the reviewers will be happy with the corrections incorporated. In the revised manuscript, the questions raised by reviewers’ have been addressed and the changes are made in Red Colour throughout the revised manuscript.

REVIEWER  1

Comment No. 1: The present manuscript have presented multiple mechanisms for PGPR enhancing plants tolerance to the drought stress, so the title of the manuscript should be changed, but not only focusing on the antioxidant mechanism.

Response: We would like to state that even though the present work is focused on the multiple mechanisms for PGPR enhancing a plant’s tolerance to drought stress, the study’s conclusion has related most of the mechanisms of PGPR directly or indirectly correlated to the antioxidant mechanism. Hence, we would like to retain the same title, which suits the Journal’s special issue. Further, another reviewer has not raised a comment on the title and hence, we would be happy to retain the same.

Comment No. 2: In the section 2 “Plant growth promoting rhizobacteria mediated drought stress tolerance in plants”, on one hand, I think the introduction for every part involved is very redundant, which should be simplified. On the other hand, siderphores producing PGPR and underlying mechanisms of the interaction with plants of should be elaborated.

Response: As per the Reviewer’s suggestion, section 2 of the revised manuscript has been updated accordingly. In addition, the revised manuscript has updated a subsection (Lines 370-382) on the importance of siderophore production by PGPR and its underlying mechanisms.

Comment No. 3: I think the architecture should be reorganized according to the correlations among multiple mechanisms, because there existed overlapped description.

Response: We agree with the Reviewer’s comments on the organization of MS. As you noted, there are several mechanisms that coexist directly or indirectly, and we would like to emphasize that utmost care has been taken in organizing the MS with all possible correlations.

Comment No. 4: In the line 90, immune responses? I think it should be modified, because there is not any immune response in the following description.

Response: As per the Reviewer’s suggestion, the sentence has been modified accordingly in the revised manuscript.

Comment No. 5: Line 128-129, what is relationship between ROS and plants’ high relative water content? Is there any published reference?

Response: As per the Reviewer’s suggestion, the relationship between ROS and plants’ high relative water content has been incorporated with proper citations in the revised manuscript.

Comment No. 6: Line 136-139, the relative references should be cited. In addition, the relationship between the production of phytohormones and high water content should be elaborated.

Response: As per the Reviewer’s suggestion, the literature on the relationship between the production of phytohormones and high water content is added and elaborated in the revised manuscript (Line 131-141).

Comment No. 7: There are so many minor errors in the manuscript. If the bacteria genus name showed repeatedly, the name should be abbreviated, and all of the species name should be italic. In addition, line 339 “2R,3R-” should also be italic. The similar errors should be modified through the whole manuscript.

Response: As per the Reviewer’s suggestion, all the binomial nomenclature cited has been italicized throughout the revised MS. The 2R,3R-butanediol has also been italicized (Line No. 341). Further, all the syntax errors have been corrected throughout the revised MS.

Comment No. 8: What does the sentence in line 406-407 mean? It should be more precise for convenient reading.

Response: As per the Reviewer’s suggestion, the sentence in lines 420-423 has been modified accordingly for convenient reading in the revised manuscript.

Comment No. 9: There are so many grammar errors in the whole manuscript.

Response: As per the Reviewer’s suggestion, the entire manuscript has been cross-checked for grammar errors and updated accordingly in the revised manuscript.

Reviewer 2 Report

Gowtham et al. reviewed the mechanism and application of PGPR in improving drought resistance in plants. The authors also put forward their own views on the current trends and development direction of PGPR research in the field of plant drought resistance. Overall, the manuscript was well organized, and  has good reference and guidance for similar research in this field. I have only one modification proposal as follows.

Abstract: The authors should not state too many plant drought stress backgrounds, but should highlight the role of PGPR.

Author Response

Reviewer 2

Authors’ response to the Reviewer(s) comments

Comment No. 1: Gowtham et al. reviewed the mechanism and application of PGPR in improving drought resistance in plants. The authors also put forward their own views on the current trends and development direction of PGPR research in the field of plant drought resistance. Overall, the manuscript was well organized, and  has good reference and guidance for similar research in this field. I have only one modification proposal as follows.

Response: We profusely thank the reviewers for their constructive comments. Herewith we are submitting the revised manuscript following incorporation of all the suggestions and hope the reviewers will be happy with the corrections incorporated. In the revised manuscript, the questions raised by reviewers’ have been addressed and the changes are made in Red Colour throughout the revised manuscript.

Comment No. 2: Abstract: The authors should not state too many plant drought stress backgrounds, but should highlight the role of PGPR.

 Response: As per the Reviewer’s suggestion, the role of PGPR has been highlighted in the abstract of the revised manuscript.